# Outcomes from a One-Week Adapted Sport and Adapted Adventure Recovery Programme for Military Personnel

**DOI:** 10.3390/sports7060135

**Published:** 2019-05-31

**Authors:** Suzanne M. Peacock, Jim McKenna, David Carless, Carlton Cooke

**Affiliations:** 1Carnegie School of Sport, Leeds Beckett University, Leeds LS6 3QS, UK; J.McKenna@leedsbeckett.ac.uk (J.M.); D.Carless@leedsbeckett.ac.uk (D.C.); 2School of Social and Health Sciences, Trinity University, Leeds LS6 3QS, UK; c.cooke@leedstrinity.ac.uk

**Keywords:** adventure, armed forces, mental health, physical activity, recovery, soldiers

## Abstract

Background: The Battle Back Centre offers a bespoke, Self Determination Theory-oriented adapted sport and adventurous training programme centred on experiential learning and reflection to support the recovery of military personnel. Aim: To identify the short-term impact of participation in the programme on positive mental health and psychological need satisfaction. Method: Participants were 978 wounded, injured and sick (WIS) personnel classified as: Wounded (battle casualties), Injured (non-battle casualties) and Sick (mental/physical illness). Participants completed the Basic Need Satisfaction in General Scale (Gagné, 2003) and Warwick and Edinburgh Mental Well Being Scale (Tennant et al. 2006) on arrival and course completion. Results: All measures of positive mental health and psychological need satisfaction showed statistically significant increases, with a large effect size, from baseline to course completion (mean ± SD change in positive mental health, competence, autonomy and relatedness were 7.19 ± 9.61, 0.46 ± 0.9, 0.27 ± 0.84, 0.26 ± 0.86, respectively, *p* < 0.05). While the average magnitude of the intervention effect for positive mental health (16%) is comparable or greater than other reported interventions, changes were achieved in a shorter time. Conclusion: Findings highlight the positive short-term effect adapted sport and adventurous activities have for WIS personnel. Declaration of interest: Work supported by The Royal British Legion.

## 1. Introduction

Deployment to a war zone places military personnel in highly stressful and dangerous situations, which can have significant consequences. Since 2006, more than 2200 UK Service personnel have been wounded in action in Afghanistan, returning home with severe wounds, illnesses, and/or disabilities [1]. Equally, almost 3000 deployed personnel were assessed as having a mental health disorder in 2014/15 [2], with as many as 13% meeting the criteria for post-traumatic stress disorder (PTSD) [3,4,5]. 

Although the experiences of deployed personnel are challenging and unique, there is also a health need among military personnel who are not battle casualties. For instance, more than 5500 UK personnel have sustained non-battle related injuries and diseases in Afghanistan since 2006 [1] the rate of mental health problems among non-deployed personnel has increased by 63% since 2007 [2]. Further, although rates of PTSD and adjustment disorders are higher for deployed personnel, rates of mood disorders are significantly higher in non-deployed personnel [2]. The challenges faced by these individuals are extensive and diverse, with recovery often being associated with a sense of helplessness and loss of self-identity [6,7] less satisfaction with life [8], reduced psychological need satisfaction [9,10], increased aggression [11] and difficulties with post-deployment social functioning and transitioning into civilian living [12,13]. Consequently, the after care of Service personnel is of great interest and importance to the Armed Forces, Mental Health Services and Service Charities because personnel who leave the Armed Forces symptomatic generally remain symptomatic and are at greater risk of social exclusion and hardship [14]. Further, upon medical discharge, the responsibility for health care is passed to the National Health Service (NHS) and Service Charities. However, NHS treatment compares unfavourably with treatment in the US’ and the appropriateness of the treatment for ex-Service personnel has been questioned [15]. 

Extensive efforts are being made to support the recovery of Service personnel. However, while interventions focus on prevention, identification and management of injury and illness, there remains limited evidence of programme outcomes [16]. Moreover, clinical models of recovery dominate military rehabilitation, focussing on symptom reduction and physical functioning [17,18]. Although this approach is necessary and can be enhanced by technological and medical advancements, this approach alone is questionable for two reasons.

First, clinical approaches often do not offer a holistic approach to recovery. In contrast, patient-centred models of rehabilitation recognise that Service personnel are trying to make sense of, and search for meaning within, their individual circumstances. Frequently, individuals are attempting to establish a new (often non-military) identity within their post-injured/ill body and integrating this into their lives [18]. This is reflected in the complete mental health model [19], which proposes mental illness and well-being are distinct, with recovery being possible amidst on-going symptoms. Second, clinical approaches to recovery can create stigmatisation. Within the UK militaries, fear of being perceived as ‘weak’ continues to be the most common stigmatising belief across time [20], with personnel preferring to ‘handle it on my own’ than seek help [21]. Instead, personnel favoured informal support, with the majority seeking help through a spouse, friend, chaplain, or internet search [22]. In light of the dominance of clinical approaches and the associated stigmatisation, Dustin and colleagues [23] (p. 329) highlight the need to explore alternative approaches to recovery outside conventional practice which ‘are not associated with hospitals, rehabilitation centres, or other clinical settings’. One area which has attracted recent attention is sport and adventurous physical activity, which is thought to compliment mainstream practices by ‘facilitating a faster return to healthy levels of psychological functioning’ [24]. 

There are a number of initiatives which currently use sport and physical activity as a tool for recovery in the USA (Wounded Warriors Programme, Disabled Sports USA), Canada (Soldier On), Australia (Australian Defence Force Paralympic Sport Program) and the UK (Battle Back Programme). However, little evidence has been generated to document programme outcomes and the impact on mental health and recovery. Moreover, existing research has mostly focussed on US populations and primarily considers combat-veterans, with small samples. In the first systematic review of its kind, Caddick and Smith [25] highlighted the therapeutic value of sport and physical activity on the subjective and psychological well-being of combat-veterans. In addition to generating positive emotions, the researchers concluded that sport and/or physical activity has the potential to shape the personal growth of combat veterans following acquisition of a disability or psychological trauma.

Therefore, recognising (i) the need to treat mental health problems during Service; (ii) the stigmatisation associated with professional medical help and preference to seek informal sources of support; (iii) the recommendation to use holistic, non-clinical approaches to recovery; (v) the role of sport and outdoor pyshical activity in recovery; (v) the dominance of US combat veterans in samples and; (vi) the need to rigorously document programme outcomes, the current study explores the role of a five-day adaptive sport and adventurous training (AS & AAT) programme in the recovery of UK wounded, injured and sick (WIS) in-Service personnel.

## 2. Method

### 2.1. Setting: The Battle Back Centre

Established in 2011, the Battle Back Centre (BBC) aims to assist the recovery of UK wounded (battle casualties), injured (non-battle causalities) and sick (mental/physical illness) (WIS) in-Service personnel. To achieve this, civilian coaches deliver the Multi Activity Course (MAC); a five-day bespoke, Self Determination Theory (SDT)-oriented programme (Ryan & Deci, 2000), using AS & AAT as a vehicle for personal development. Providing 24× MAC per year, the facility operates on a participant:coach ratio of 3:1, with each course offering space to 22 Service personnel across the three Services.

Following an optional morning walk and breakfast, each day opens with a daily brief where personnel are informally introduced to various psychological concepts or strategies (e.g., motivation, attitude, goal setting). Following this, personnel participate in a variety of AAT (e.g., indoor rock climbing, mountain biking, kayaking) and AS (e.g., archery, wheelchair basketball, seated volleyball, indoor bowling) activities. The presence of a full-time Technical Advisor and an extensive array of bespoke adaptive equipment facilitates enables all participants, regardless of individual circumstance, to participate in activities and experience success. Each day concludes with a review that encourages personal reflection and discussion at a group and individual level. This process encourages individuals to extract meaning from their experience and apply their learning to other life domains. Following a communal meal, social activities are held in the evening to promote social interaction and integration, including a cinema trip, quiz night, interest talks and evening walks.

### 2.2. Participants and Recruitment

Participants were WIS personnel directed to attend the BBC by their chain of command. Formal inclusion criteria are that participants will be (i) male and female UK Service personnel (British Army, Naval Service, Royal Air Force), including mobilised reserves; (ii) either wounded, injured and/or sick and; (ii) be independently mobile and self-medicating. Due to the nature of the participants and the sensitivities within this population, the researchers were advised to reduce participant burden and increase participant anonymity. To achieve this, we were recommended to avoid the disclosure of personal information that could lead to non-participation. Therefore, specific demographic and background information is not reported. However, in line with the demographic profile of the Armed Forces (Defence Analytical Services and Advice [DASA], 2014), the sample was dominated by white male Army personnel under the age of 25 years. Although attendance at the BBC is mandatory for Army personnel and recommended for the Royal Air Force and Naval Service, participation in the evaluation was voluntary. On arrival at the Centre, personnel were introduced to the aims of the evaluation and invited to participate. Voluntary written consent was then collected.

### 2.3. Data Collection

Approved by the Leeds Beckett University Ethics Committee, a formative service evaluation has documented the Centre’s development across the 11 Pilots and its progression into full operational capacity. Initial qualitative, open-ended feedback from those attending the pilot courses suggested that the programme generated experiences, reflection and learning that were consistent with mechanisms of change associated with the Self-Determination Theory [26]. Therefore, Pilot 10 marked a change in the evaluation strategy with the initiation of a pre–post assessment using validated questionnaires, which explored possible changes to the elements of self-determination and psychological wellbeing. Specifically, volunteers completed two self-report measures on arrival and completion of the BBC Course. This study focusses on data collected during 69 MACs over a 25-month period between March 2012 and April 2015.

### 2.4. Measurements

Psychological Needs Satisfaction. The Basic Psychological Needs Scale is a family of scales which measure need satisfaction in specific domains (i.e., work, interpersonal relations, physical education) and life as a whole. The 21-item Basic Need Satisfaction in General Scale (BNSG-S) [27] was included as a measure of both outcome and of treatment fidelity, and was purposefully selected for this population because it is context-free and addresses WIS personnel’s need for autonomy (7 items; e.g., ‘I feel like I am free to decide for myself how to live my life’), need for competence (6 items; ‘Often, I do not feel very competent’ reversed) and need relatedness (8 items; ‘I get along with people I come into contact with’) within life in general. This approach is thought to provide insight into the general recovery of this military population. Responses were based on a 7-point Likert scale ranging from 1 (‘not all true’) to 7 (‘very true’). After reversing the scores of nine negatively worded items, average scores for the three sub-scales were calculated, with high scores representing greater need satisfaction. The scale has reported Cronbach alpha coefficients of 0.69, 0.71, and 0.86 for the autonomy, competence, and relatedness scores, respectively [27].

Positive Mental Health. The Warwick–Edinburgh Mental Well-Being Scale (WEMWBS) was developed from the Affectometer-2 [28] and covers two dimensions of wellbeing: (i) hedonic perspective and (ii) eudaimonic perspective. The 14-items represent the only valid, positively worded scale to measure positive mental health. Responses are based on a 5-point Likert scale ranging from 1 (‘none of the time’) to 5 (‘all of the time’). Answers are summed to provide an overall score, ranging from 14 (minimum) to 70 (maximum). The scale is a user-friendly and psychometrically sound measure of mental wellbeing, providing high internal consistency (α = 0.89) and good reliability [29]. It provides a normal distribution with no floor or ceiling effects [30] and is responsive to change [31], making it an appropriate tool for monitoring mental wellbeing in population samples. Furthermore, it is suitable for individuals aged 13+ years [30], making it appropriate for the current setting where average reading ages are low [32].

### 2.5. Data Analysis

Data analysis was conducted using the IBM SPSS Statistics 22 package (Version 22, IBP Corporation, Armonk, NY, USA). All data were cleaned, screened for missing values and assessed for normality. With instances of missing values, the baseline score was brought forward [33,34], thus representing no change to psychological wellbeing. This conservative approach assumed a neutral intervention effect of the BBC.

Descriptive statistics, together with changes in scores and significance values were identified. To assess the impact of participation in the MAC, paired sample *t*-tests were conducted on each of the dependent variables (positive mental health, autonomy, competence and relatedness). The significance level was set at 0.05. However, because statistical significance is likely to be achieved due to the large sample size [35], the effect size will also be presented to establish the magnitude of the intervention effect [36]. This will be reported through Cohen’s d, with 0.2 representing a small effect, 0.5 representing a moderate effect and 0.8 representing a large effect [37].

## 3. Results

In total, 1020 WIS Service personnel attended 69 MACs across a 25-month period. Of these, 96% (*n* = 978) volunteered to participate in the research. From this group, 971 participants completed the WEMWBS, with 15% of these (*n* = 177) providing WEMWBS data at only one time point. A total of 957 participants completed the BNSG-S, with 21% (*n* = 200) of these participants providing BNSG-S data at only one time point.

### Overall Changes to Positive Mental Health and Psychological Need Satisfaction

Descriptive statistics, together with changes in scores and significance values are presented in Table 1. All measures of positive mental health and psychological need satisfaction showed statistically significant increases from baseline to course completion, each with a large or moderate effect size (Table 1). The largest increase was shown in positive mental health, 16% (7.19 ± 9.61, t (970) = −23.332, *p* < 0.0005, two-tailed, eta = 0.44). Of the psychological needs met during the week, competence increased the most, increasing by 11%, followed by a 6% rise in autonomy and a 5% rise in relatedness.

## 4. Discussion

Responding to the need for military recovery interventions to document programme outcomes [16], the present study evaluated the impact of a bespoke five-day AS & AAT programme on the well-being of in-Service UK WIS personnel. Moreover, this study moves beyond research addressing combat veterans with small sample sizes (e.g., Caddick & Smith [25]) and explores the health needs among a very large military sample that includes all categories of in-Service WIS personnel: the full panoply of Service employees. Findings suggest the innovative, SDT-orientated programme produced significant positive changes. Personnel reported significant increases to positive mental health and satisfaction of three core psychological needs.

While the WEMWBS is commonly used as a measurement of positive mental health at the population level [30] and has been used extensively in interventions with the general population that often last 4–12 weeks [38,39,40,41,42], this study marks the introduction of the WEMWBS as viable outcome measures for military interventions. While the scale of the effect of the current programme is comparable to or greater than existing interventions, it was achieved in a much shorter time frame (only five days). With more than a quarter of UK Service personnel experiencing common mental health disorders [3,4], the development of positive mental health within this population is essential because it embraces more than the absence of ill-health. Embracing the idea that positive mental health is ‘a state of complete physical, mental and social well-being’, suggests that WIS personnel experience mental well-being even alongside mental illness.

Underpinning the facilitation of these positive mental health experiences is the Self-Determination Theory (SDT) [26]. SDT suggests that optimal psychological functioning, growth and integrity are only achieved with the satisfaction of innate psychological needs for (i) autonomy, (ii) competence and (iii) relatedness. Even though performance outcomes are not the primary focus of delivery, perceived competence experienced the largest significant increase (11%). While military life is underpinned by physical activity, recovery is often associated with prolonged periods of (frequently enforced) inactivity following injury/illness [43]. Moreover, this is often accompanied by reduced perceived competence and belief in personal capability [8,44,45,46]. Therefore, the flexible physicality of the programme seemed to appeal to many participants, perhaps leading to this increased competence outcome. Previously, the MAC was shown to provide personnel the opportunity to ‘do things again’ and promotes participation in both familiar and new sporting activities [44,45,46]. A similar notion of ‘discovery’ was reported by four injured combat veterans following a nine-day climbing expedition of Mount Kilimanjaro [24].

A further explanation for the rise in competence score may be due to the combined effects of completing a physical challenge during the day, followed by the retelling of the experience during end-of-day reflections and being encouraged to consider how these (re)discoveries could apply to daily life. Experiencing success, in addition to participating alongside other WIS personnel, enabled MAC participants to reappraise and develop an accurate representation of their physical ability following injury [43]. These findings are consistent with previous research, which emphasised both focussing on ability, rather than disability, and providing opportunities for comparison and reappraisal. Both featured during (i) a National Veterans Wheelchair Games (NVWG) and Winter Sports Clinic (WSC) [47]; (ii) a climbing expedition to Mount Kilimanjaro for injured combat veterans [24]; and (iii) three-day U.S. Paralympic Military Sport Camp (USPMSC) [48]. 

Autonomy also increased significantly, rising by 6%. Within a military context, this development is particularly significant because the Armed Forces have been described as a ‘dependency culture’ [49]. However, while sport has been suggested to enhance autonomy by assisting combat veterans to cope with the psychological consequence of acquiring a disability [24], or to take control of their behaviour [50], developments to autonomy are only occasionally recorded. For instance, although competence and relatedness were facilitated during a three-day USPMSC, expressions of autonomy were limited [48]. Accounting for this, Hawkins et al. suggested that the limited evidence for autonomy was most likely due to attendance being recommended or required by their chain of command. However, this was also true for MAC participants. Therefore, recognising the importance of an autonomy-supportive environment for predicting psychological need satisfaction [51], it could be argued that course content and delivery facilitated enhanced autonomy. While Hawkins et al., did not detail programme delivery, the MAC operates on the ethos ‘challenge-by-choice’, which encourages individuals to take charge of their BBC experience. This ownership may be influential in the development of overall autonomy scores.

Finally, relatedness also increased significantly, rising by 5%. Three explanations might account for this rise. First, is the occurrence of a positive reappraisal of existing relationships that occurred during the course. Previous research has documented how some personnel came to value relationships with their families and/or partners as a result of reflection that took place during the course [45]. Second, is the shared experience of military culture. Military culture is distinctly different from civilian organisations or institutions, with its own language, humour, rituals and beliefs [52]. Personnel often believe civilians do not and cannot understand their military experiences [3,53]. However, the MAC has previously demonstrated its ability to enable personnel to unite through a shared military experience and rekindle elements of the military culture which were often lost as a result of the isolation associated with injury and/or illness [43].

Third, is the shared experience of injury, illness and/or recovery. Recognising that personnel can often feel unique, embarrassed and/or isolated with their injury/illness [44], the MAC may offer a normalising and reassuring experience. Previous research conducted at the Centre suggests personnel can easily interact with and observe other WIS participants at varying stages of their recovery process and compare themselves and offer support [43]. This process enables participants to share similar authentic and first-hand experiences, and offer support, insight and suggestions to other group members, while also providing perspective, hope and inspiration [54]. Of 11 studies which were included in a systematic review, eight acknowledged the role of sport and physical activity in the facilitation of social well-being [25]. For instance, consistent with outcomes from the MAC, participation in a three-day military sport camp also enhanced relatedness through the shared experience of the military and traumatic injury [48]. 

Moving beyond the single programme, the findings also have wider implications. Clinical models of rehabilitation dominate military health care, with levels of physical functioning or symptomology often acting as markers of recovery. However, with the appropriateness of current NHS recovery programmes for veterans being questioned [17,49], alongside the tendency for clinical approaches to predominantly help the minority [55], there is a need to explore alternative avenues for enhancing the recovery of WIS Service personnel. Considering the holistic definition of recovery [56,57,58,59] and patient models of rehabilitation [18], an alternative approach could be through sport and/or physical activity. These are perceived as ‘normal’, ‘healthy’ behaviours and not accompanied by the stigma or side effects associated with counselling and medication [58]. Furthermore, recognising the ideology of a soldier and the emotional inexpressiveness of some men, it is suggested that these men need assistive activities to stimulate personal exploration and emotional expression [59]. While these findings provide promising evidence to support the role of AS & AAT in the recovery of in-Service UK WIS personnel, further studies are necessary to explore several areas. First, it remains unclear whether attendance at the MAC creates a halo effect, with participants potentially experiencing positive improvement that declines after the course. Changes to positive mental health and psychological need satisfaction require exploration in the weeks, months and years following participation in the programme. Second, it may be useful for future research to collect demographic information (i.e., gender, rank, age) and differentiate between categories of wounded, injured and sick to determine effects across sub-populations. Third, while the magnitude of change is greater than many existing non-military interventions, further studies are needed to determine how the outcomes compare to other existing five-day military residential programmes. Finally, incorporating in-depth qualitative research will add depth to the research, providing detailed insight to ‘the Battle Back experience’ and the underlying mechanisms which generate the documented changes in wellbeing. 

## 5. Conclusions

This study marks a progression in the recovery literature, documenting the role of a five-day AS & AAT programme in the promotion of well-being in UK in-Service WIS personnel. Findings suggest attendance of the MAC generates statistically significant improvements to positive mental health and satisfaction of psychological needs. Moreover, while the scale of the effect is comparable to, and often greater than, existing interventions, it is achieved in a much shorter time frame. Future research should explore the longer term implications of attending the MAC.

## Figures and Tables

**Table 1 sports-07-00135-t001:** Descriptive statistics, reliability coefficients and change in scores of positive mental health and basic psychological need satisfaction.

		Time 1		Time 2		Change in Score				
Measure and Variable	*n*	Mean (SD)	α	Mean (SD)	α	Mean (SD)	%	95% CI	*p* value	Cohen’s d (Effect Size)
Positive Mental Health	971	45.05 (11.37)	0.95	52.24 (10.3)	0.95	7.19 (9.61)	15.9	−7.8 to −6.59	0.000	1.5 (large)
Autonomy	957	4.41 (0.98)	0.72	4.63 (0.93)	0.72	0.27 (0.84)	6.1	−0.32 to −0.22	0.000	0.67 (large)
Competence	957	4.26 (.97)	0.65	4.73 (.96)	0.68	0.46 (0.90)	10.87	−0.53 to −0.41	0.000	1.03 (large)
Relatedness	957	4.93 (1.03)	0.67	5.19 (0.97)	0.66	0.26 (0.86)	5.27	−0.21 to −0.20	0.000	0.59 (moderate)

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
