# Peer review of "Outcomes from a One-Week Adapted Sport and Adapted Adventure Recovery Programme for Military Personnel"

_sports, 2019, doi:10.3390/sports7060135_

Round 1
Reviewer 1 Report
Overall, I think this is a well-written paper but there are areas to work on. I have some questions/comments that follow:
- You have used cumulative data from 2012-2015. How can you justify that these data waves can be combined and considered as one pool of data? You need to provide enough justification that data from different years are comparable.
- In the Participants section, you should provide adequate descriptive information about your sample such as age range, gender, the percentage of wounded, injured and sick for each sample and the total sample. This allows readers to have a better understanding of your data.
- I believe you should report the reliability estimates of your measures under the Measurement section and not in Table 1.
- In the Results section, you mentioned that 177 participants provided WEMWBS response only to one time point and 200 participants provided BNSG-S response only to one time point. This means that you had 801 (i.e., 978-177) complete data for WEMWBS and 778 (i.e., 978-200) complete data for the BNSG-S. This leaves us with 601 (i.e., 978-[200+177]) complete data on both measures. The sample sizes in Table 1 do not reflect these findings. For example, the sample size for the positive mental health was reported as 971 and I couldn't find any information supporting this number. The same thing for a sample size of 957 for other measures.
- In line 180 p.4 you reported the t-test for the positive mental health as 7.19+ 9.61 (the same thing happens all across the results section). At first, I had no idea what is the 9.61 number. But after looking at Tables I realized that it is the standard deviation of change scores. In this case, it should be reported as 7.19 ± 9.61. Please correct this for the other reported values as well.- Why did you choose eta as a measure of effect size? You could simply report Cohen'd which is more intuitive. You need to justify your choices.
- In the tables and the text, always put 0 before the decimal. For example, instead of .36 report 0.36.
The reported confidence interval for the positive mental health in Table 1 spans zero. Is this a typo?
- I strongly believe that the whole section (i.e., section 3.2) on comparing WEMWBS adds no value to your manuscript. You are comparing apples to oranges and the focus of your study is not assessing the usefulness of this tool. You just used it in your study and you can reference all supporting evidence of its applicability and usefulness under the Measurement section.
Here are my suggestions:
- Remove that part about the WEMWBS and spend more time analyzing your data.
- You seem to have plenty of data so instead of using simple paired samples t-test, utilize other information such as gender and classification of participants as wounded, injured and sick as the grouping variable and employ ANCOVA and Factorial ANOVA for a better understanding of the impact of BBC
- Then you can ground your findings and discussions based on the Self-Determination theory and other related mental health theories.
Reviewer 2 Report
In the current manuscript, Peacock et al. have systematically measured the positive mental and psychological benefits that stem from the participation of sport and adventurous training program. To promote the recovery of military personnel, this program emphasized experiential learning and reflection. The sample size is impressive with almost one thousand participants. The positive outcomes were measured carefully, using both the Basic Need Satisfaction in General Scale and Warwick and Edinburgh Mental Well Being Scale. Thus, the present manuscript represents a significant step toward better addressing the mental health needs of military personnel.
Please move the detailed statistical results from the abstract section to the results section.
The title, while objective, fails to convey the major finding of the current study.
Round 2
Reviewer 1 Report
- If the confidence interval of the PMH in Table 1 spans zero then the p-value cannot be 0.000. A confidence interval that spans zero mean non-significant difference. Your confidence interval for the PMH contradicts your p-value and effect size.
- The sample sizes reported in Table 1 do not make sense given the information you provided in terms of responses at each time point. You should justify it.
Author Response
Please find the responses to the reviewer below:
- I apologise for the typo within the table, there was a negative sign missing. The confidence interval is -7.8 to -6.59 meaning it does not span zero.
- I have updated the sample size to make it clearer.
Regards,
Suzanne